# Neuro-Nutraceutical Polyphenols: How Far Are We?

**DOI:** 10.3390/antiox12030539

**Published:** 2023-02-21

**Authors:** Maria Teresa Gentile, Iolanda Camerino, Loredana Ciarmiello, Pasqualina Woodrow, Lidia Muscariello, Ida De Chiara, Severina Pacifico

**Affiliations:** Department of Environmental Biological and Pharmaceutical Sciences and Technologies, University of Campania “Luigi Vanvitelli”, 81100 Caserta, Italy

**Keywords:** central nervous system, neurodegenerative disorders, polyphenols, dyshomeostasis microbiota, gut–brain axis

## Abstract

The brain, composed of billions of neurons, is a complex network of interacting dynamical systems controlling all body functions. Neurons are the building blocks of the nervous system and their impairment of their functions could result in neurodegenerative disorders. Accumulating evidence shows an increase of brain-affecting disorders, still today characterized by poor therapeutic options. There is a strong urgency to find new alternative strategies to prevent progressive neuronal loss. Polyphenols, a wide family of plant compounds with an equally wide range of biological activities, are suitable candidates to counteract chronic degenerative disease in the central nervous system. Herein, we will review their role in human healthcare and highlight their: antioxidant activities in reactive oxygen species-producing neurodegenerative pathologies; putative role as anti-acetylcholinesterase inhibitors; and protective activity in Alzheimer’s disease by preventing Aβ aggregation and tau hyperphosphorylation. Moreover, the pathology of these multifactorial diseases is also characterized by metal dyshomeostasis, specifically copper (Cu), zinc (Zn), and iron (Fe), most important for cellular function. In this scenario, polyphenols’ action as natural chelators is also discussed. Furthermore, the critical importance of the role exerted by polyphenols on microbiota is assumed, since there is a growing body of evidence for the role of the intestinal microbiota in the gut–brain axis, giving new opportunities to study molecular mechanisms and to find novel strategies in neurological diseases.

## 1. Introduction

The brain is the most complex organ in the human body; it receives information, and coordinates and influences all the activities of the body from physical motion to the secretion of hormones, from the creation of memories to the sensation of emotion. The human brain uses around 20 percent of our body’s total energy, and consists of around 100 billion neurons and 1000 billion glial cells [1]. Brain cells are tightly connected, and communication problems in one area can disrupt other brain activity, meaning brain disorders can cause widespread problems. There are many diseases and illnesses that can affect the brain, of which the most complicated are chronic degenerative diseases, which range from neurodegenerative diseases, such as Alzheimer’s (AD) or Parkinson’s disease (PD), to tumors. They are devastating, can substantially and adversely affect an individual’s quality of life, and are associated with high costs for patients, their families, and society as a whole [2].

In the last hundred years, thanks to the progress in medical science, human life expectancy has significantly increased. However, advancing age is often associated with a greater sensitivity to chronic organ diseases due to compromised metabolic and immune functions with a peculiar impact on the central nervous system (CNS) [3]. As such, with the global ageing population, the prevalence of diseases associated to the decline brain functioning worldwide is estimated to double every 20 years, and expected to increase to 115 million affected individuals by 2050. Among these different forms of dementia, AD will be the most prevalent [4]. Currently, all approved pharmacological treatments for most forms of dementia act only on the symptoms without having an impact on the prevention of their onset [5]. Thus, there is an urgent need to explore new and more efficacious therapeutic approaches focused on dietary phytochemicals, mainly polyphenols, acquirable through diet or nutraceutical supplementation. These compounds could represent important resources in the discovery of drug candidates against dementia or valuable neuroprotective supplements since they are recognized to be antioxidant, anti-amyloidogenic, anti-inflammatory, and anti-apoptotic.

## 2. Chemistry and Sources of Polyphenols, a Sustainable Strategy to Support Central Nervous System (CNS) Functions

Polyphenols are a large family of specialized plant compounds, which share at least two phenolic rings held together by joining structural elements. In the natural product chemistry scenario, polyphenols are biosynthetically derived from the shikimate and/or acetate/malonate pathways, yielding a large structural variability, encompassing a wide range of compounds [6]. In fact, based on their chemical characteristics, polyphenols can be classified into the two main classes of flavonoids and non-flavonoids [7]. These compounds are distributed diversely in plants, with some metabolites being highly ubiquitous.

Food plants and derived foodstuffs enjoy an ever-increasing recognition as a source of these compounds, which could be favorably obtained from other undervalued sources such as medicinal plants, commonly used to obtain essential oils, and agro-wastes (Figure 1). Flavonoids are among the specialized metabolites of greatest interest as nutraceuticals for prevention and prophylaxis. These plant pigments, characterized by a 2-phenylchromonic base skeleton, can be differentiated into subclasses based on the oxidation state of the heterocyclic ring. Closely related to flavonoids are neoflavonoids and isoflavonoids, the latter reported as chemoprotective and an alternative therapy for a wide range of hormonal disorders [8,9]. Non-flavonoid polyphenols enjoy a much wider structural variability and among these, stilbenes, such as the well-known resveratrol, diarylheptanoids, such as curcumin, lignans with precious phytoestrogenic activity, and xanthones deserve increasing attention. All these compounds are extractable from natural sources, mostly food, to the extent that they are described as dietary phytochemicals, rarely as aglycones, more commonly in the form of glycosides or acyl- or prenyl-derivatives. Furthermore, tannins also feasibly affect human health; cereals, tea, and fruits such as bananas, apples, and grapes are edible sources of these polymeric compounds which, based on their ability to be hydrolyzed in simple phenols, are distinguishable in hydrolysable tannins (HTs), and condensed tannins (CTs). Phlorotannins, deriving from phloroglucinol polymerization, are suggested new ingredients in food. These polymers are mainly from microalgae and seaweeds [10].

Although the food sources of polyphenols have been extensively investigated, still today a thorough knowledge of the polyphenolic heritage in plants is not available and the use of more performing extraction techniques and advanced structural elucidation techniques allows a daily update in knowledge [6]. The abundance in fruit, vegetables, cereals, and beverages makes them bioactive non-nutrients with an average consumption of about 1 g/day per person [11]. It is certain that these compounds are biosynthesized as a chemical defense strategy, intrinsically implemented by plants to counteract environmental adversity [12,13]. The production of reactive oxygen species (ROS) and nitrogen ones (RNS), due to abiotic stresses (e.g., drought or UV rays’ exposure), forced the decrease of some biochemical processes related to plant growth/development, augmenting the biosynthesis of polyphenols. Indeed, polyphenols play a far superior role in plant ecology. In plant tissues, their effects range from the release and suppression of growth hormones to UV shields to protect against ionizing radiation and to provide coloration, form herbivore deterrence to the prevention of microbial infection, and the influence of signaling molecules in maturation and other process growth [14]. In human nutrition, polyphenols are consumed in all diets and, together with micronutrients such as vitamins and minerals, they are stored or temporarily retained in the body to facilitate basic biochemical processes and protect against stress, helping to improve long-term health in many different ways [15,16].

The first evidence of the beneficial role of polyphenols in human health came from surveys in the 1960s and 1970s, while further epidemiological studies have indicated that the intake of polyphenols may be associated with a reduced risk of developing cancer, cardiovascular disease, and neurodegenerative disorders [17,18,19,20,21]. As in plants, in animals and humans the antioxidant activity of polyphenols is thought to be a valuable aid in the inhibition of pathophysiological perturbations in the redox circuits. The ability of polyphenols to prevent or slow down oxidative stress-related diseases, such as cancer and neurodegeneration, appears to be due to their intrinsic ability to neutralize, deactivate, or suppress ROS and RNS by transferring an electron or hydrogen atom or by acting as inhibitors of lipoperoxidation. Furthermore, counteracting the overproduction of ROS species, pure polyphenols and/or plant complexes enriched in polyphenols have been shown to modulate inflammation cellular signaling, to interfere with gene regulation, and to regulate gut microbiota. These biological features are in line with the hypothesis of utilizing polyphenols as valuable candidates to maintain the brain and to counterbalance the pro-oxidant mixture, consisting of polyunsaturated fatty acids, a high transition metals content, and ascorbate in the central nervous system [22]. Polyphenols have a well-known effect on ROS protection by acting as scavengers, or regulating the expression of several genes coding for enzymes involved in stress oxidative protection, such as increasing complex 1 expression and activity [23]. Furthermore, polyphenols play an important role in cell death inhibition acting on some proteins involved in apoptosis such as Bcl/Bax, caspase 3, and protein kinases [24]. Epidemiological studies and associated meta-analyses strongly suggest that the long-term consumption of diets rich in plant polyphenols offers protection against the development of non-communicable diseases [25], beneficially affecting human brain function [26]. Furthermore, the bidirectional brain–gut interactions, in which the intestinal microbiome is the protagonist, increase the attention on food polyphenols, since the microbiota, which is an individual entity, processes these compounds from the diet, transforming them into metabolites with health benefits [27]. This has led to relentless in vitro, preclinical, and clinical research aimed at highlighting the role of polyphenols for the prevention and treatment of neurodegenerative diseases, in particular AD and PD, but the low in vivo bioavailability appears to be the limiting factor.

## 3. Polyphenols and Microbiota–Gut–Brain Axis

The growing awareness of a bidirectional communication between the brain and gut, and of the role of the intestinal microbiota in the gut–brain axis, has opened new scenarios for the study of the molecular mechanisms underlying these neurological diseases and the discovery of a new generation of disease modifying strategies.

Polyphenols bioactivity depends on absorption rate, metabolism, and bioavailability and is regulated by the interaction with other dietary nutrients, such as proteins, carbohydrates, fibers, and lipids. In addition, the biological activity of the derivative metabolites may differ upon their fermentation by gut microbiota [28]. In particular, the majority of polyphenols, due to their structural complexity, are characterized by low bioavailability, and after ingestion are recognized as xenobiotics. Therefore, complex polyphenols are fragmented into low-molecular-weight and readily absorbable metabolites by the gut microbial community (Figure 2). On the contrary, small polyphenols can be directly adsorbed in the small intestine [29]. Most of the natural flavonoids, for instance, exist in the form of glycosides which are easily metabolized into aglycone and more easily absorbed by the body by the intestinal microbiota [30].

Highly concentrated flavonoids are directly absorbed by the body, exerting their antioxidant and anti-inflammatory effects [31]. At the same time, microorganisms in the intestinal tract can transform flavonoids into small molecular substances, such as phenolic acids [32]. A great number of studies in the last decade focused the attention on the microbial community involved in flavonoid metabolism. *Actinobacteria*, *Firmicutes*, *Bacteroidetes*, and *Proteobacteria* are the four bacterial phyla involved in flavonoid conversion, particularly *Firmicutes* families in the intestinal microbiota, including *Lactobacillaceae*, *Streptococcaceae*, *Enterococcaceae*, and *Clostridiaceae* [33]. *Streptococcus*, *Lactobacillus*, and *Eubacter* are able to metabolize epicatechin and procyanidin B1 in 5-(30,40-dihydroxyphenyl)-valerate lactone, after their ingestion in the human body [34]. *Lactobacillus* L-2, *Bifidobacterium* B-9, and *Rhizobacteria* JY-6, convert quercetin into 3,4-dihydroxyphenylacetic acid, 3,4-dihydroxybenzoic acid, 4-hydroxybenzoic acid, 3(3-hydroxybenzoic acid), propionic acid, and other small molecules which can be absorbed and utilized by the body [35]. Keranmu and co-authors [36] have recently studied the role of intestinal microbiota in the biotransformation of liquiritigenin, an important dihydroflavonoid compound found in *Glycyrrhiza uralensis*, that has a wide range of pharmacological properties, such as antitumor, antiulcer, anti-inflammatory, and anti-AIDS effects. They demonstrate that gut microbiota are able to metabolize liquiritigenin in vitro into three main metabolites: phloretic acid, resorcinol, and a probable davidigenin, which may play an important role in the biological effects of this flavonoid, including effects on neurodegenerative diseases and disorders [37].

On the other hand, dietary polyphenols affect microbiota composition and functions by acting on their central metabolism [38]. Today, polyphenols are considered to have a prebiotic action, stimulating the growth of beneficial microbiota while inhibiting harmful bacteria [39]. Indeed, a variation in the daily intake of polyphenols may lead to differences in the metabolites of phenolics. Additionally, a variation of gut microbiota composition affects the bioavailability and bioactive effect of polyphenols and other metabolites [40]. The decrease of *Bifidobacterium* spp. and the increase in proteobacteria lead, for example, to a deregulation of cholesterol levels and to GABA dysmetabolism. Furthermore, a decrease in bacterial short-chain fatty acids (SCFAs)-producing bacteria, such as *Roseburia intestinalis* and *Faecalibacterium prausnitzii,* was observed in PD individuals, whereas their gut microbiota was abundant in *Akkermansia muciniphila*, *Alistipes shahii*, *Alistipes obese*, *Alistipes ihumii,* and *Candidatus gastranaerophilales*, and depleted in *Prevotella*, *Lactobacillus,* and *Streptococcus* spp. Bacteria. *Escherichia coli* abundance was clearly associated with PD severity, as it enhances intestinal hyperpermeability, endotoxins translocation into the colonic mucosa, inflammation, and amyloidogenesis [41]. Every individual has their own microbiota composition, and this may lead to differences in polyphenols metabolism; thus, a similar daily intake of polyphenols can have different healthy effects on people with differences in their microbiota composition. This is important since changes in the composition of microbiota may contribute to the onset of neurodegenerative disorders that increase with age.

It is well known that adults undergo dramatic changes in their microbiota composition following diet modifications. Furthermore, ageing is related to specific changes in microbiota diversity which result in health outcomes in the elderly. Due to their “prebiotic-like” effect, polyphenols modify gut microbiota composition, inhibiting or increasing the growth of specific bacteria. Rastmanesh [42] observed that the ingestion of different polyphenols results in a different ratio between the healthy/non-healthy gut, favoring the growth of *Bacteroides* that possess a higher number of glycan-degrading enzymes.

In vitro and in vivo studies showed that different polyphenols can modulate the growth of specific bacterial strains. Indeed, gut microbial enzymes can transform, through dihydroxylation, methylation, decarboxylation, or non-covalent and covalent bond breakage, phenolic compounds into small molecular phenolic compounds (e.g., gallic acid, protocatechuic acid, and p-coumaric acid) which can stimulate the growth of beneficial microbiota and inhibit harmful bacteria. For instance, polyphenols can increase beneficial strains such as Bifidobacteria and Lactobacilli, reducing the number of pathogens, such as *Clostridium perfringens* and *Clostridium histolyticum* [43]. Dong and co-workers [44] proposed that polyphenols and their metabolites from carrot could potentially promote gut health by acting like a prebiotic and modulating the fecal microbe composition in a positive manner. They observed, in fact, that polyphenols and their metabolites from carrots inhibited the growth of pathogenic bacteria such as *Clostridium* spp. And stimulated health promoting bacteria belonging to *Escherichia*, *Lactococcus,* and *Bifidobacterium* genera. Similarly, Sun and co-authors [45] observed that polyphenols from oolong tea, green tea, and black tea samples significantly influenced the intestinal microbiota via increasing the *Bifidobacterium* spp., *Lactobacillus* spp., and *Enterococcus* spp. A recent study investigated the influence of phenolic compound extracts from three colored rice cultivars on gut microbiota. After microbiota fermentation, the major metabolites were protocatechuic acid, chlorogenic acid, caffeic acid, and *p*-coumaric acid. The presence of these metabolites promoted the presence of *Prevotella*, *Megamonas,* and *Bifidobacterium*, while the presence of *Escherichia-Shigella* was inhibited. Moreover, the authors observed changes in short-chain fatty acids (SCFAs), revealing that phenolic compounds generated more propionate and acetate but not butyrate [46].

Evaluating the effect of punicalagin from *Punica granatum* L. on insulin resistance, it was shown that polyphenols hindered the IKKβ/NF-κB inflammatory pathway by regulating gut microbiota homeostasis and upregulating liver autophagy activity [47]. Cyanidin-3-*O*-glucoside, which was rather stable in the stomach and intestine, and gut microbiota-convertible in protocatechuic, vanillic, and *p*-coumaric acids, increased *Muribaculaceae* bacteria, while decreasing *Lachnospiraceae* bacteria, suggesting a positive impact of the anthocyanin supplementation on body weights, glucose/lipid metabolism, and inflammatory markers [48]. The beneficial effects of cyanidin-3-*O*-glucoside, whose edible sources are different fruit berries and drupes, among which *Vitis aestivalis* L., *Sambucus nigra* L., and *Prunus avium* L., could be due to its claimed antioxidant potential. Recently, this compound was shown to activate an Nrf2-antioxidant response element, and to protect vs. glutamate-induced oxidative and endoplasmic reticulum stress in HT22 hippocampal neuronal cells [49]. Gut microbiota modulation by condensed tannins is in line with their protection against inflammatory injury in the intestinal barrier, and their ability to fight dysbiosis in murine models, otherwise preventing intestinal inflammation, permeability, and oxidative stress [50]. Where polyphenol bioavailability is low, the synthesis of nanocomposites such as Res-selenium-peptide, containing the bioactive resveratrol, appears to be a strategy to pursue [51]. In particular, the first data suggest that Res-selenium-peptide improved gut microbiota, further exerting valuable antioxidant activity, reducing intracellular ROS, and promoting enzymatic activity, the inhibition of Aβ aggregation, and downregulation of Aβ-induced neuroinflammation via the NF-κB/MAPK/Akt signaling pathway [52].

It is becoming clear that dietary polyphenols through their metabolites contribute to the maintenance of gut health by the modulation of the gut microbial balance. The literature is enriched daily with data proving how the effect of polyphenols on the intestinal microbiota is closely related to the antioxidant and anti-inflammatory action of these substances. Since brain dysfunctions are associated with dysbiosis of the gut microbiota, regulation of microbiota composition using polyphenols, or other probiotics and prebiotics, may help to restore gut equilibrium and to set up new therapeutic interventions in neuropathologies, and may result in a partial or complete reversion of these diseases [53]. Due to the diversity of natural products, the complexity of metabolic activities of human intestinal bacteria, and other factors, research on the metabolism and transformation of natural products by human intestinal bacteria is still at an early stage and further understanding of the interactive regulation of active ingredients and intestinal microbiota are needed. We are still far from defining doses and times of administration, but the positive evidence also on healthy subjects [54] opens up investigative scenarios in which nutrition enriched with polyphenolic nutraceuticals is a useful weapon against neurodegenerative brain disorders.

## 4. Polyphenols: Antioxidants for Brain Health

Dietary polyphenols exert pleiotropic effects, and their preventive role against the development of chronic-degenerative pathologies is often associated with their ability to act as antioxidants [55,56]. These exogenous compounds are thought to restore the oxidative imbalance that leads to biomolecule damage and disturbed or disrupted redox signaling (oxidative distress). The brain is particularly susceptible to oxidative stress, which leads to mitochondrial respiratory chain dysfunction, and is implicated in neuroinflammation and neuronal death [57,58]. The neuroinflammatory response is regulated by immunocytes (such as microglia, astrocytes, and peripheral immune cells), cytokines, and chemokines [59]. Microglia are the resident immune cells of the central nervous system. Under normal circumstances, resting microglia play a critical role in maintaining tissue homeostasis and promoting brain development [60]. Following a variety of pathological stimuli, including infections, brain damage, stroke, and neurodegeneration, microglia secrete high levels of activation of proinflammatory factors, and an excessive accumulation of these factors causes neuronal damage [61].

Thus, the use of natural antioxidant compounds such as polyphenols has been suggested as a valuable therapeutic strategy in the neurodegenerative process, so much so that by counteracting ROS overproduction, polyphenols could interrupt a series of processes which, in the diversity of neurodegenerative pathologies, cause ROS as a species to feed and self-feed the neuronal damage. In the following paragraphs, the latest evidence on the role of even little-known polyphenols as neuroprotective agents will be mainly discussed, with evidence suggesting that the overproduction of ROS is implicated in β-amyloid peptide neurotoxicity. The exposure of cultured neurons or neuronal cell lines to Aβ is able to increase the intracellular levels of ROS leading to NF-κB activation [35]. Furthermore, it has been observed that the activation of microglial cells, associated with neuroinflammation, determines an increase in the levels of acetylcholinesterase (AchE) and further promotes the genesis of free radicals [62]. To provide other examples, alterations in pro- and antioxidant compounds have been reported in the postmortem tissue of PD-affected individuals. Furthermore, an increase in total iron was found in the substantia nigra of PD patients. Iron could increase oxidative stress by promoting the formation of hydroxyl radicals from H_2_O_2_ via the Fenton reaction [63]. Reductions in glutathione (GSH) levels in the substantia nigra have also been reported [64]. Alterations in GSH levels are an early event, therefore a reduction could promote or be a consequence of oxidative stress. Since GSH is involved in H_2_O_2_ detoxification, any reductions could result from increased concentrations of H_2_O_2_ and the highly reactive hydroxyl radicals in the presence of metals. Lipoperoxidation and oxidative DNA damage occurrence further support the involvement of oxidative stress in PD onset and perpetuation [65]. The increase in ROS production, due to an increased turnover of dopamine, may not be controlled by the activity of antioxidant enzymes resulting in a compensatory hyperactivity of dopaminergic neurons which may become self-destructive [66]. Another index of PD oxidative stress could be the excessive increase of NF-κB in the nuclei of dopaminergic neurons of the substantia nigra which contributes to increased intracellular ROS levels [67].

Polyphenols play their antioxidant protection role principally via activation of several protein kinases signaling molecular pathways such as Keap1/Nrf-2/ARE [68] (Figure 3). The interaction with Keap1 leads to Keap1/Nrf2 complex disruption. Therefore, Nrf2 can translocate to the nucleus, where it binds antioxidant response (ARE) elements, adenylate and uridylate (AU)-rich, and enhances the expression of genes coding for antioxidant proteins such as heme oxygenase-1 [69]. Polyphenols’ neuroprotective role is carried out by activating the pathways regulating transcription, translation, proliferation, and growth, such as MAPKs (mitogen-activated protein kinase). These bind tyrosine receptor kinases (Trk), activating protein kinase cascades and cAMP (3′,5′-cyclic adenosine monophosphate) response element-binding protein (CREB), increasing the expression of Bcl-2, Bcl-xL [70]. Moreover, this binding with Trk receptors promote neurons, hippocampal neuronal cells, and spinal ganglion neurons survival by activating the downstream signaling cascades [71]. Polyphenols downregulate pro-inflammatory transcription factors such as NF-κB [72].

## 5. Polyphenols vs. Neurodegenerative Disease

Neurodegenerative diseases are a heterogeneous group of age-related disorders, for which there is a slow but irreversible deterioration of brain functions [73]. Alzheimer’s disease, Parkinson’s disease, Huntington’s disease (HD), amyotrophic lateral sclerosis (ALS), frontotemporal dementia (FTD), and spinocerebellar ataxias are just a few examples of these diseases, which mostly differ in their pathophysiology, so much so that some affect memory and cognition and others the person’s ability to move, speak, and breathe [74]. PD is characterized by a distinct ageing-independent loss of dopaminergic neurons in the substantia nigra pars compacta (SNpc) region and a decrease in dopamine levels. This leads to motor and non-motor phenotypes linked with intraneuronal protein aggregates, the Lewy bodies, which are from the gut to the neocortex, and shows α-synuclein as the main constituent [75]. PD is the second most widespread neurodegenerative disorder worldwide after AD [76], which in turn represents the most common type of dementia.

The term ‘dementia’, also known as ‘major neurocognitive disorder’, is not one specific disease but rather a group of symptoms that happen because of a disease [77]. Thus, dementia could be defined as an age-related irreversible condition resulting in a progressive cognitive decline that reduces a person’s ability to perform daily activities. The National Institute of Health classifies, beyond AD, other different forms of dementia such as vascular dementia (VaD), dementia with Lewy bodies (DLB), frontotemporal dementia (FTD), and mixed dementias [78]. Actually, a rare form of dementia, namely young-onset dementia, accounts for approximately 3% of dementia cases, and occurs in people under the age of 65 [79].

The World Health Organization (WHO) estimates that the number of individuals with dementia worldwide is approximately 55 million, with this number expected to reach approximately 78 million by 2030 and 139 million by 2050 [World Health Organization. Geneva, Switzerland: World Health Organization; 2021. Fact sheets of dementia https://www.who.int/news-room/fact-sheets/detail/dementia, accessed on 20 September 2022.].

Strategies to reduce or prevent AD, including diet and nutrition, have been broadly investigated, but the multiple etiologies still do not allow a therapeutic answer to be found. In fact, AD has multiple etiological factors: genetics, environmental factors, and in general, lifestyle. Genetic and non-genetic AD risk factors have been identified, but their role in the onset and/or progression of neuronal degeneration remains elusive. Analogously, different hypotheses have been proposed (Figure 4) and on their basis some drugs are proposed.

Awareness of the occurrence of impairment of the cholinergic system firstly favored the identification of acetyl-cholinesterase (AChE) inhibitors such as galantamine, an amaryllidaceous alkaloid, isolated from various species of the Amaryllidaceae, including daffodils (*Narcissus pseudonarcissus* L.) and snowdrops (*Galanthus* spp., *Leucojum* spp.). Galantamine is a competitive AChE inhibitor, launched to the market in the European Union and the United States in 2001 [82,83]. Galantamine shows a sensitizing action on nicotinic acetylcholine receptors in the central nervous system, and is less toxic than the pyrroloindole alkaloid physostigmine from *Physostigma venenosum* Balfour. Physostigmine analogues, such as rivastigmine, are also used to treat mild to moderate PD-associated dementia. Indeed, this symptomatic treatment is able to compensate, without stopping, the progressive and inevitable loss of neurons, and the identification of the complex mechanisms underlying the neurodegeneration are still being investigated [84]. Although it is clear that the cholinergic system is not the only one involved in pathogenesis, the search for new compounds with anti-acetylcholinesterase activity is still topical, and recently the flavonols myricetin, quercetin, and fisetin from the *Phyllanthus emblica* Linn. fruit were found to reversibly inhibit AChE, quenching the AChE fluorescence and modifying enzyme secondary structures [85].

AD is a chronic disorder that slowly destroys neurons, causing serious cognitive disability, primarily due to senile plaques and neurofibrillary tangles mainly composed of amyloid β peptides (Aβ), the former, and hyperphosphorylated Tau protein, the latter. Aβ accumulation and Tau hyperphosphorylation represent typical hallmarks of several forms of dementia and the prevention of their formation could be a targeting opportunity in the therapy of neurodegenerative diseases [86]. In particular, to counteract these hallmarks of the pathology, and considering that oxidative stress is recognized as one of the earliest events in AD and appears to play a central role in Aβ generation, neuroinflammation, and neuronal apoptosis, a strong antioxidant species could be a strategy of choice. Indeed, this may be able to interrupt the vicious cycle of Aβ generation and oxidation. Numerous mechanisms have been proposed for explaining how polyphenols positively act. It was observed that supplementing rodents and humans with isolated polyphenols may improve memory and cognition [87,88] and that these compounds exert their beneficial effects either through their ability to lower oxidative stress and inflammation or by directly altering the signaling involved in neuronal communication, calcium buffering ability, neuroprotective stress shock proteins, plasticity, and stress signaling pathways. Several findings demonstrate that some polyphenols are able to inhibit Aβ aggregation and Tau phosphorylation. Nevertheless, the effective brain uptake of polyphenols is still regarded with some reservations and the true mechanisms by which they may permeate the blood–brain barrier (BBB) are not fully understood [89].

### 5.1. Polyphenols Inhibit Amyloid Genesis

Numerous mechanisms have been proposed for explaining the positive anti-amyloidogenic impact of naturally occurring polyphenols, and the recent literature is devoted to clarifying, through the application of powerful spectroscopic and spectrometric techniques, their anti-AD effectiveness at the molecular level. Ono et al. [90] reported that wine-related flavonoids (myricetin, morin, quercetin, kaempferol, (+)-catechin, and (−)-epicatechin) inhibited the formation of Aβ fibrils from fresh Aβ40 and Aβ42 and destabilized preformed Aβ fibrils in a dose-dependent manner. The ability of polyphenols to slow down and actively counteract oxidative stress processes, thus preventing Aβ polymerization, was proposed to be mediated by their direct interaction with Aβ or by their binding to ions, which facilitate Aβ aggregation [91]. Similar effects of curcumin, rosmarinic acid, tannic acids, epigallocatechin gallate (EGCG), olive tree-extracted polyphenols, and resveratrol were identified [92,93,94,95,96,97,98].

Most of the evidence, even the most recent, derives from computational studies or studies which, exploiting the fluorescence of the dye thioflavin T (ThT), quantify the formation and inhibition of amyloid fibrils. The use of cellular models, such as differentiated SH-SY5Y cells, is also one of the most adopted strategies, while not fully reproducing the disease features when these cells are exposed to toxic Aβ oligomers leading to neurodegeneration. It appears to be clear that the determination of the anti-radical efficacy and the reducing power, as well as the ability to inhibit the enzymatic activity of the secretases, are also involved in preliminary investigations, likewise to identify a structure–activity relationship. An inverse relationship between antioxidant activity and Aβ aggregation was recently established for some pure polyphenols or their phytoceutical mixture [99].

Polyphenols can act on an Aβ monomer preventing its fibrillation via monomer stabilization or oligomer formation. This is probably due to the polyphenols–metal ions interaction which is able to promote Aβ aggregation [100]. Indeed, thanks to their ability to interact with the β-sheet structure, polyphenols can also disaggregate oligomers and fibrils. The aglycone structure can be fundamental in exercise activity, while a moderate impact is due to glyconic moiety. Examining morin and isoquercitrin flavonols and two flavanones, such as hesperidin and neohesperidin, it was found that the flavonols were able to exert a greater antioxidant activity, to attenuate caspase-3 and -9 activation, and to protect against glutamate-induced oxytosis [101]. The flavonol quercetin and its rutinosyl derivative were both capable of inhibiting the formation of Aβ fibrils and disaggregated Aβ fibrils, as well as affecting the activity of the β-secretase enzyme (BACE). Investigating the behavior of these compounds through a cellular system overexpressing the Swedish APP mutation (APPswe), which provides oxidative stress for increased Aβ production, it was observed that both quercetin and rutin inhibited ROS genesis, increasing the intracellular GSH content and favoring a decrease in the lipid peroxidation index compared to control APPswe cells [102]. On the other hand, not all the compounds analyzed in vitro seem to have promising properties. In this regard, through a combination of in silico and in vitro methods, a library of 180 flavonoids, previously investigated as inhibitors of BACE1, the rate-limiting enzyme in Aβ production, was analyzed, and dihydromyricetin, taxifolin, and tamarixetin were revealed as agents capable of inhibiting the growth of fibrils and, with the exception of dihydromyricetin, Aβ oligomers [103]. Molecular docking analyses and molecular dynamics simulations highlighted the ability of amentoflavone-type biflavonoids to promote the breakdown of Aβ fibrils, as also revealed by the thioflavin T fluorescence assay, and that the ability was enhanced when the substitution of hydroxyl groups capable of hydrogen bond formation at two positions on the biflavonoid scaffold occurred [104].

Based on in vitro data, several in vivo studies have been carried out, and animal models have been realized as a valuable, although perfectible, tool. In this context, some polyphenols such as EGCG, resveratrol, and curcumin decreased amyloid levels and plaque formation in mice brains. The diarylheptanoid curcumin, in particular, is known to disrupt Aβ fibril formation and to reduce AD pathology in mouse models, defining, as investigated through solid-state NMR spectroscopy, structural modifications of Aβ(1-42) aggregates in the Asp-23-Lys-28 salt bridge region and near the C terminus [105]. However, due to the poor aqueous solubility and bioavailability of curcumin, its clinical applications are limited, and to increase the bioavailability of curcumin, a highly stable and completely water soluble system consisting of a polymeric nanoparticle-encapsulated curcumin conjugate with gold nanoparticles decorated on the surface was formulated and found to inhibit Aβ(1–16) aggregation [106]. The treatment of Aβ42-induced SH-SY5Y cells with curcumin and piperine highlighted that the combined approach could serve against Aβ via the modulation of various mechanistic pathways. In fact, high-throughput microarray profiling on the extracted RNA evidenced alterations in the expression profiles of genes in the neuronal cell model, and ITPR1, GSK3B, PPP3CC, ERN1, APH1A, CYCS, and CALM2 were found to be putative genes involved in AD pathogenesis [107]. Positive data were also found from epigallocatechin-3-gallate (EGCG), one of the main green tea constituents, which was found to be biologically able to penetrate the blood–brain barrier and inhibit the fibrillation of amyloid proteins [108]. The long-term oral consumption of EGCG at a relatively high dose (15 mg/kg) ameliorated memory impairment [109], reducing Aβ cytotoxicity. ^15^N and ^1^H dark-state exchange saturation transfer (DEST), relaxation, and chemical shift projection NMR analyses suggested that EGCG interferes with secondary nucleation events known to generate toxic Aβ assemblies [110]. The anti-amyloidogenic and anti-inflammatory effects following the oral administration of EGCG (50 mg/kg) for 4 months in APP/PS1 transgenic mice, consisted of the improvement of dendritic integrity and expression levels of synaptic proteins, and in the alleviation of microglia activation, decreasing pro-inflammatory cytokines (IL-1β) and increasing anti-inflammatory cytokines (IL-10, IL-13) [111]. Furthermore, using an APP/PS1 mice model and orally supplementing them for 15 months with EGCG and/or FA (30 mg/kg each), an amelioration of brain parenchymal and cerebral vascular β-amyloid deposits and decreased abundance of amyloid β-proteins compared with either EGCG or FA single treatment was observed. This could be due to the complementary anti-amyloidogenic properties of EGCG, which act as an α-secretase activator, and ferulic acid, a β-secretase modulator. The co-treatment also ameliorated neuroinflammation, oxidative stress, and synaptotoxicity [112].

Other interesting compounds are oleuropein aglycone and oleocanthal from extra virgin olive oil and olive leaf [113], which are also shown to effectively counteract amyloid aggregation and toxicity [114]. In particular, the phenylethanoid oleocanthal was shown to enhance the Aβ clearance across the blood–brain barrier, decreasing amyloid load in the hippocampal parenchyma and microvessels [115]. The inhibitory activity against Aβ(1-42) aggregation of catechol-type and non-catechol-type flavonoids was further studied through a ^1^H–^15^N heteronuclear multiple quantum coherence (HMQC) analysis and mass spectrometry [116]. Phenolic groups of flavonoids, through their H-bond formation ability, suppress amyloid mature fibrils destabilizing β-sheet structures of Aβ peptide [117].

Furthermore, the potential anti-amyloidogenic effects from plant extracts enriched in polyphenolic antioxidants have also been broadly examined. It was reported, for instance, that *Curcuma longa* extract, besides acting as an antioxidant, significantly increased anti-inflammatory cytokine IL-4 production and reduced Aβ and tau levels in Aβ overexpressing mice [118]. The neuroprotective properties of the *Ginkgo biloba* extract EGb761, broadly investigated in murine models and humans, is attributed to both flavonols and terpene-lactones. Among individual EGb761 components, kaempferol and quercetin provided maximum oxidative stress attenuation in Alzheimer’s disease models [119]. Aqueous garlic extracts showed anti-amyloidogenic properties inhibiting Aβ fibril formation and also defibrillating Aβ preformed fibrils [120], whereas berry polyphenol extract, whose digested derived fraction was shown to cross BBB, mediated inflammation and cell survival signaling pathways, enhancing neuroplasticity, neurotransmission, and calcium buffering [121]. Meganatural-Az^®^GSPE, a nutraceutical grape seed polyphenolic extract, was demonstrated to be bioavailable at the organism levels and to reduce aggregation of Aβ peptides into high molecular weight Aβ oligomeric species in the brain of a transgenic AD mouse model [122,123,124]. The ability of phenol-rich extracts from *Laurus nobilis* leaves to promote a dose-dependent regression of the formation of Aβ oligomers in neuronal SH-SY5Y and SK-N-BE(2)C cell lines, pretreated in the previous 24 h with the Aβ(25-35) neurotoxic fragment, expounded a promising neuroprotective effect [123]. Analogously, an alcoholic extract from deterpenated *Pistacia lentiscus* L. leaves was investigated for its protective effects vs. Aβ(25-35)-induced toxicity in SK-N-BE(2)-C cells [124]. Moreover, a sage leaf phenol extract, accounting for almost 50% of abietane diterpenes, 40% of phenylpropanoid constituents, and 10% of flavonoids, was found to exert anti-lipoperoxidative and antioxidant properties, and inhibit the AChE enzyme in SH-SY5Y cells far greater than donepezil [125].

The main pathological hallmarks of the sporadic and familial forms of the disease are a predominant and progressive degeneration of the dopaminergic neurons (SNpc) associated with a systematic progressive iron accumulation, leading to a dopamine depletion in the striatum, disappearance of neuromelanin, and appearance of intracellular Lewy bodies with the major component consisting of aggregated α-synuclein.

The anti-amyloidogenic properties of polyphenols may also benefit Parkinson’s neurodegeneration, since the deposition of Lewy bodies, consisting of a large part of α-synuclein aggregates, in neuron cytoplasm is a key pathological PD hallmark [126]. In fact, the progressive degeneration of dopaminergic neurons of the substantia nigra pars compacta, together with iron accumulation, provides dopamine depletion in the striatum, thus neuromelanin disappear and Lewy bodies become visible. α-Synuclein, intrinsically disordered and in a physiological α-helix conformation, forms oligomers to then give amyloid fibrils. Polymorphic structures have been observed that could be rod or twister, laying new foundations for understanding the pathology and developing therapies. Recent data have observed that the nucleus of polymorphic rod fibrils is composed of residues 37–99, and that of polymorphic twister fibrils is from residues 43–83, while the α-synuclein segment 44–47 appears to be involved in the elongation of the amyloid fibril. As α-synuclein amyloid aggregation occurs through the induction of a protein phase transition, stopping this process could be synonymous with hope. Recently, curcumin, which was known to effectively block α-synuclein aggregation in vitro, underwent an extensive study aimed at exploring its ability to interact with α-synuclein during phase separation. The data acquired stated that curcumin effectively interacts with the protein hydrophobic regions, further decreasing its fluidity within the condensates, and thus inhibiting amyloid genesis. Amyloid inhibition by curcumin also extends to those from α-synuclein E46K and H50Q mutants [127]. The anti-amyloidogenic effects of EGCG also occur at the level of a-synuclein, since EGCG binds directly to α-synuclein, as well as Aβ, promoting their assembly into large non-toxic spherical oligomers. The main effect of polyphenols is due to the interaction with the C-terminal domain of a-synuclein. This reduces the conformational plasticity of the protein and its ability to be converted into fibrils [127]. Dopaminergic cell death and α-synuclein aggregation are shown to be strictly connected [128].

### 5.2. Polyphenols Inhibit tau Hyperphosphorylation

The intraneuronal neurofibrillary tangles, the main constituent of which is the tau (tubulin-associated unit) protein is another target in anti-AD research. Tau proteins are amply expressed in the axons of CNS neurons, and also in the somatodendritic compartment of neurons, oligodendrocytes, and non-neural tissues [86]. Tau proteins undergo post-translational modifications, mainly consisting of *O*-glycosylation, and phosphorylation. The latter modifies the protein shape affecting its ability to bind tubulin and promote microtubule assembly. Abnormal folding of the microtubule-associated protein tau leads to the aggregation of tau into paired helical filaments (PHFs) and neurofibrillary tangles. The phosphorylation degree reflects aberrant activity of protein kinases and phosphatases. The tau hyperphosphorylation and the formation of β-sheet-rich fibrils that ultimately contribute to the synaptic insufficiency, neuronal death, and cognitive decline observed in AD patients may be exacerbated by disturbed copper homeostasis [129]. The copper disequilibrium could be detrimental to neuronal survival, favoring an increase in the expression of the p53 upregulated modulator of apoptosis and the upregulation of nucleoside diphosphate kinase NME1. It was observed that quercetin acts as an antioxidant when administered in mild oxidative stress conditions, involving PI3K/Akt and ERK1/2 signaling, while it is pro-oxidant in severely damaged neurons.

The amyloid hypothesis stated that the Aβ of various lengths are by the APP proteolytic cleavage, which first involves β-secretase (BACE1), followed by γ-secretase. The transmembrane aspartic protease β-site APP cleaving enzyme 1 (BACE1) has been identified as the essential β-secretase, which is a major determinant that predisposes the brain to Aβ amyloidogenesis. This enzyme is in high amounts in the brain with sporadic onset AD, and an its increase in mild cognitive impairment (MCI) in the brain suggests that it is an early AD indicator.

Oligostilbenes such as ampelopsin C and vitisin A from *Vitis thunbergii* var. taiwaniana exerted a BACE1 inhibitor-like activity, decreased sAPPβ levels, and increased sAPPα levels. They have the ability to inhibit β-secretase activity in N2aWT cells and in SH-SY5Y cells [130]. In the same phytochemical investigation, stenophyllol B was isolated, but it appeared to reduce Aβ generation by enhancing α-secretase activity through APP non-amyloidogenic processing [130]. The chemical structures of bioactive oligostilbenes are shown in Figure 5.

Anti-amyloidogenic polyphenols interfere with tau aggregation [131], inhibiting tau phosphorylation [132]. Grape seed polyphenol extract (GSPE) reduces tau pathology in the TMHT mouse model of tauopathy [133]. Transmission electron microscopy revealed that GSPE induced profound dose- and time-dependent alterations in the morphology of PHFs with partial disintegration of filaments. The GSPE mechanism may include a noncovalent interaction of polyphenols with proline residues in the proline-rich domain of tau. The neuroprotective effects of polyphenols could be exerted through the modulating signaling cascade that controls neural apoptosis. In particular, current findings suggest that several polyphenols impact on the ERK pathway, increasing ERK phosphorylation and supporting neural plasticity and survival. It has been found that luteolin increases neurite outgrowth and expression of GAP-43, a neuronal differentiation marker in PC-12 cells and that this effect could be blocked pharmacologically by ERK1/2 inhibition [134]. Catechin, procyanidin A1, and procyanidin A2 from the seed of *Litchi chinensis* Sonn. inhibit hyperphosphorylated tau via the upregulation of IRS-1/PI3K/Akt and downregulation of GSK-3β. Molecular docking results further demonstrate that these polyphenols exhibit a good binding property with insulin receptors [135].

The hyperphosphorylation of tau proteins is not unique to AD. In fact, as tau is a phosphoprotein that stabilizes the neuronal cytoskeleton interacting with microtubules and nuclei [136], its genetic mutations are also directly involved in the genesis of several other neurodegenerative diseases such as frontotemporal dementia which represent a heterogeneous group of progressive neurodegenerative dementias with prominent behavioral alterations [137]. Growing evidence has ascertained that the overexpression or mutation of α-synuclein increases tau phosphorylation [138], and a potential association between tauopathy and sporadic PD was from the detection of tau aggregates and deposits in ~50% of PD brains [139]. The aggregation of neurofibrillary tangles, the abnormal hyperphosphorylation of tau protein, and the interaction between tau and α-synuclein are all conditions contributing to cell death and poor axonal transport observed in PD and parkinsonism [139].

### 5.3. Polyphenols and Metal Dyshomeostasis

Oxidative stress and mitochondrial dysfunction are associated with the vicious circle that defines neurodegeneration. The key role of mitochondria in modulating apoptosis, ferroptosis, and inflammasome activation allows them to be involved in the development and progression of neurodegenerative diseases, especially when impairment of their biogenesis and abnormal mitochondrial dynamics occur. Mitochondrial impairment also disrupts cellular Ca^2+^ homoeostasis. Transition metal ions with accessible *d*-orbitals, acting as structural and metabolic cofactors, or electron carriers are involved in oxygen transport and activation, neurotransmitter synthesis, DNA replication and transcription, cell–cell interactions, and extracellular matrix construction/breakdown. Metalloproteins such as ceruloplasmin, superoxide dismutase, hemoglobin, and cobalamin contain transition metals such as Cu, Zn, Fe, Co, and Mn. The metabolism of Cu and Zn is regulated by metallothioneins and a key role of Cu, Zn, and Fe in the formation of Aβ fibrils has been suggested [140]. In this regard, Cu interacts with both the APP protein and the Aβ42 peptide, metalloproteins capable of sequestering metal ions by triggering the toxic redox cycle. Cu and Fe increase the production of ROS, and are promoters of the genesis of hydroxyl radicals via the Fenton reaction, whereas Zn, by modulating the binding of Cu to the Aβ42 peptide, exerts an indirect action.

Since polyphenols have been described as metal chelators, their involvement in the prevention of AD and other neurodegenerative diseases should be mandatory. Several studies have focused their attention on how the structural properties of polyphenols can favor the activity, but some studies at the cellular level underline that the interaction of polyphenols–metals can mediate the establishment of a pro-oxidant environment. The dose certainly plays a fundamental role. Cyanidin, malvidin, rutin, and quercetin, tested in the range 0.05–4 mM, measuring the malondialdehyde generated during the linoleic acid oxidation Cu^2+^-exposed emulsion, revealed dual pro-oxidant and antioxidant activity [141]. The coordination of Cu(II) was observed to increase the antiradical efficacy of luteolin, while it did not impact the efficiency of apigenin [142]. Curcumin, administered at doses of 0.2 and 0.5 mg/kg of diet for 7 days, alleviated Cu^2+^ toxicity in *Drosophila melanogaster*, exerting anti-AChE activity and restoring cellular antioxidants [143]. A recent study determined the affinity and binding ratios in flavonoid-Cu(II) interactions in buffered aqueous solutions [144]. For this purpose, 18 different compounds were investigated, including the two isoflavones biochanin A and genistein (Figure 6). Their impact on Cu(II)-associated redox activity was evaluated by in vitro assays, and flavonoids with an appreciable Cu(II) binding capacity were evaluated for their ability to alter copper trafficking in cellular models. Among the selected flavonoids, the effects detected in yeast cells appeared to be different. In particular, 3-hydroflavone, a synthetic flavonol, was suggested to chelate extracellular copper without transport into the cell, luteolin can transport copper into the cell, affecting its intracellular availability, and quercetin, even if it may facilitate copper import, did not have a beneficial impact on extracytosolic copper availability.

When HepG2 cells were treated with luteolin and EGCG, the EGCG–Cu(II) complex was observed to decrease cell viability compared to EGCG alone, whereas the cell viability-lowering effect of luteolin was attenuated by the addition of Cu(II). This behavior of luteolin has been attributed to its relative hydrophilicity, which would allow it to alter the balance of reactive oxygen species (ROS) in the extracellular environment. However, it was found that quercetin, while showing no changes in intracellular copper levels or cell viability, induces a marked increase in copper chaperone for superoxide dismutase expression in the presence of Cu(II). This is consistent with the ability of quercetin not to affect copper import, but to modulate intracellular copper distribution and availability [144].

The effects of myricetin on copper-induced toxicity in SH-SY5Y cells were also recently studied, highlighting that at doses of 5 and 10 µg/mL, the flavonol, acting as a pro-oxidant, exacerbated the effects of copper and cell death [145]. Notably, myricetin was observed to promote chromatin condensation and loss of membrane integrity by upregulating PARP-1 and downregulating Bcl-2 expression. The inhibitors of PARP-1, ERK1/2, JNK, protein kinase (PKA), and voltage-gated calcium channels again promoted the toxic effects of myricetin, further reducing the survival of SH-SY5Y cells. Staurosporine prevented the toxic effects of myricetin, while inhibitors of the p53, p38, and PI3K/Akt pathways were without effect. Atomic force microscopy studies have shown that treatment with myricetin induces cell surface roughness and reduced elasticity [145].

Polyphenols act as hard Lewis bases to form complexes with hard Lewis acids, such as Fe(III). When the ortho-dihydroxyl moieties are present, the chelating ability increases. This is in line with polyphenols being able to stabilize Fe(III) more than Fe(II), the binding of polyphenols to Fe(II) reduces the reduction potential of iron, and Fe(II)-polyphenol complexes can be rapidly oxidized to Fe(III)-polyphenolic complexes [146]. Through a Fenton’s-reaction-based system, releasing ultra-weak chemiluminescence, composed by Fe^2+^, ethylene glycol-bis (β-aminoethyl ether)-N,N,N′,N′,-tetraacetic acid (EGTA), and hydrogen peroxide, the anti- and pro-oxidant activities of different polyphenols (5–50 µM) were investigated, and chemical features such as the presence of catechol or methoxyphenol appeared to be able to effectively scavenge hydroxyl radicals, thus exerting antioxidant activity, and to regenerate, acting as a pro-oxidant, Fe(II), by the reduction of Fe(III) [147]. Iron dyshomeostasis is associated with cell senescence in age-related disorders; thus, compounds that are able to counteract iron-dependent oxidation could be protective against the unrelenting cell death [148].

The available data show that the response of flavonoids, which often show a similar behavior in cell-free tests, can be considerably different and the study of the dynamics and outcomes of the interactions with the different cellular components needs to be better studied, above all better account keeping for the dose effect of these compounds.

Targeting metal dyshomeostasis is also a key issue in PD therapy [149] and knowledge of metal-binding speciation is of primary importance to predict the efficacy of compounds of interest, their ability to remove dysregulated metal ions from toxic deposits such as α-synuclein complex, and redistribute metal ions into safe deposits (conservative chelation). While the role of α-synuclein has been ascertained, which through interaction with the translocase of the outer membrane 20 (TOM20) defines mitochondrial impairment and excessive ROS production, the specific effects of metal coordination on the intrinsically disordered α-synuclein protein have only recently been investigated, finding that the protein shape is very sensitive to the binding of metals such as copper and zinc [150]. Furthermore, although the role of a-synuclein remains elusive, it was observed that the C-terminal region charge distribution of the negatively charged ions can be profoundly affected by Ca^2+^ binding which can modulate following protein–membrane interactions [151]. Both Fe(II) and Fe(III) bind to α-synuclein, favoring its oligomerization and the constitution of a β-sheet structure. This leads to increased lipoperoxidation levels, reduced GSH, and induced ferroptosis [152]. The anti-lipoperoxidant activity of polyphenols could serve to elude to ferroptosis cell death. In fact, this non-apoptotic regulated cell death that is dependent on iron and ROS and is characterized by lipid peroxidation [152].

### 5.4. Polyphenols and Neuroinflammation

Curcumin, as well as other polyphenols, based on their antioxidative effects, appeared to preserve cells against lipid peroxidation, and mitochondrial permeability transition, also exerting peculiar anti-inflammatory activity. Curcumin is a PPARγ agonist, able to fight both acute and chronic neuroinflammation [153,154]. In this latter context, it was observed to increase hippocampal neurogenesis and BDNF/Trkb/PI3K/Akt signaling [153]. Neuroinflammation is the activation of the brain’s innate immune system in response to an inflammatory insult and is characterized by a series of cellular and molecular changes within the brain [155]. The formation of protein aggregates and their consequent auto-aggregation within the neuronal cells are events that result from a neuroinflammatory cascade that leads to a communicative disequilibrium between glial cells and neurons following the activation of microglia and astrocytes. In AD, glial activation is from NF-kβ activation, the synthesis and release of proinflammatory cytokines such as TNF-α, IL-1, IL-6 and IL-12 affecting neuronal receptors with an overactivation of protein kinases [155]. The positive effects of polyphenols are further linked to their ability to activate SIRT1, whose protective role in neuroinflammation-related diseases has been recently reviewed [156]. Beyond resveratrol, which as an activator of SIRT1 modulates the activity of different proteins, such as the peroxisome proliferator-activated receptor-gamma coactivator (PGC-1α), FOXO proteins, NF-κB [157], and curcumin, other polyphenols are of interest. Quercetin 3-*O*-galactoside, also known as hyperoside, was recently observed to inhibit lipopolysaccharide-induced apoptosis and inflammation in HT22 cells through SIRT1 upregulation, which in turn promoted oxidative stress and neurotrophic factor reduction in the cell system [158]. The chronic treatment of old rats with a green tea polyphenol extract, named polyphenol-60, or catechin SIRT1, increased SIRT1 protein levels with benefits in cognitive function, while NF-κB hippocampal levels were unmodified [159].

### 5.5. Polyphenol Based Formulations for Enhancing Their Neuroprotective goals

Despite several promising neuroprotective properties of polyphenols, their low stability, poor bioavailability, and therefore their ability to cross the blood–brain barrier have long been debated. Similarly, questions have been raised about the ability of these compounds to arrive at adequate concentrations where they need to perform their function [89]. This has favored the optimization of different formulation strategies. In fact, although the use of in vitro transport models that mimic the passage through the BBB has repeatedly ensured that the lipophilicity of these substances is able to overcome the BBB obstacle, the possibility of preventing the structural degradation of these compounds, and also their rapid clearance, appears possible through the structuring of liposomal type encapsulates or the use of other carrier forms (e.g., lipid or polymeric nanoparticles, and micelles). The size and lamellarity of the cationic liposome have been found to be critical in the protection of trans-resveratrol from degradation [160], whereas the nasal delivery of resveratrol could be advantageously achieved when chitosan-coated lipid microparticles were used [161]. Transferrin-modified liposomes appear to be useful for overcoming the poor blood–brain barrier penetration of a promising anti-AD compound such as α-mangosteen xanthone, [162]. Similarly, co-administration with multiple lipophilic compounds, such as α-tocopherol, has proven to be a valid approach [163]. The rapid metabolism of polyphenolic compounds can be advantageously controlled through delivery systems using nanoparticles, and the potential of nanoparticle-mediated drug delivery systems to overcome barriers in the delivery of pure or blended polyphenols has been recently reviewed [164]. This is the case of the formulation of lipid nanoparticles functionalized with the RVG29 peptide, a 29 amino acid peptide derived from rabies virus glycoprotein, and loaded with quercetin [165]. Recently, curcumin, whose poor pharmacokinetic profile largely compromises its therapeutic potential [166], has been biostabilized and its cerebral permeability has been increased by loading into chitosan-coated niosomes [167]. The latter are non-ionic surfactant-based vesicles, which can move through the cerebral endothelium [168]. The use of bioactive extracts as a basis for structuring drug delivery systems has also been considered. Solid lipid nanoparticles loaded with both dopamine and grape seed proanthocyanidin extract have been found to be attractive candidates for non-invasive delivery from the nose to the brain [169]. Indeed, the synergistic effect in a plant matrix could also be effective for delivering bioactive compounds. In order to better mimic the common reactions in the gastrointestinal tract, the study of the digestion mechanisms of blackberry polyphenols and the preparation of a fraction after deglycosylation has verified, through the use of a simplified in vitro BBB model, that the transport of quercetin and myricetin, as such and as their glycosides, and kaempferol was feasible, whereas the transport of ellagic acid was not detected [89]. However, where ellagic acid was loaded into a nanoparticle system, its release allowed aluminum chloride-induced neurotoxicity to be effectively counteracted in the brain of rats [170].

## 6. Conclusions

Polyphenols are considered useful agents for preventing neurodegenerative disorders and an interesting treatment opportunity, often yet to be validated. The broad distribution of these compounds in plants and plant-based dietary foodstuffs, together with their great and fascinating structural variability open a wide possibility of actions that go far beyond the discussed and acclaimed antioxidant activity.

In vitro studies and in vivo investigations, mainly carried out on animal models, suggest that polyphenols could be able to eradicate and reverse important phases of AD and PD onset, or even to counteract neuroinflammation.

Therapeutic use still suffers from limitations often related to the poor solubility and bioavailability of these specialized metabolites, and there has been a growing interest in the exploitation of nano-delivery systems, with the aim of optimizing stable and effective formulas. Indeed, if some polyphenols, among the more ubiquitous ones such as flavonolic glycosides, or those particularly bioactive and with proven multi-target action, such as resveratrol and curcumin, have gained a lot of attention, the neurobioactivity of many other polyphenols are relegated to a low level of knowledge.

A multidisciplinary approach that combines the nutraceutical chemical aspects of edible plants and that deeply obtains insight into the bioactivity of fractions enriched in polyphenols, or of pure compounds, is necessary for their full usability.

## Figures and Tables

**Figure 1 antioxidants-12-00539-f001:**
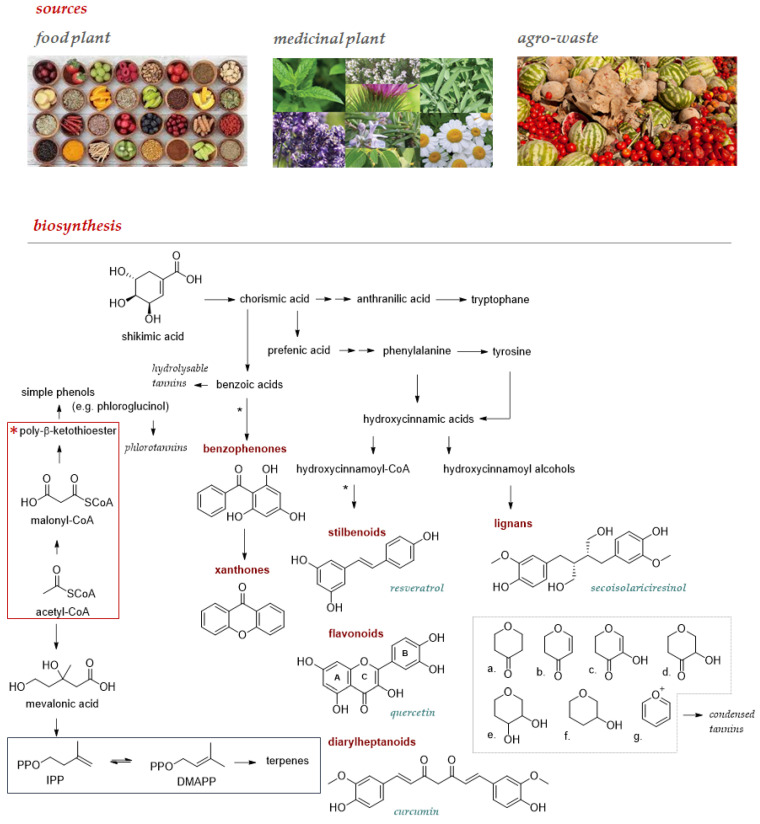
Main sources to be used to extract and/or isolate pure polyphenol compounds, whose structural variability and complexity are from the interweaving of intermediates from the three key pathways of secondary metabolism. The metabolic pathway of shikimic acid and that of acetate/malonate allow both to obtain simple phenols, but their combination opens up the biosynthesis of benzophenones, xanthones, or still stilbenoids, flavonoids, and diaryleptanoids (*). Lignans are polyphenols derived from the oxidative coupling of cinnamyl alcohols that are formed in the shikimic acid pathway alone. The prenylation observed in some bioactive polyphenols exploits intermediates of the mevalonic acid pathway. The precursors of the prenylated compounds, isopentenyl diphosphate (IPP) and dimethylallyl diphosphate (DMAPP), are in the blue frame. In the grey box, the C nucleus of the flavonoid skeleton is given attention since its oxidation degree allows: a. flavonones; b. flavones; c. flavonols; d. dihydroflavonols; e. leukoanthocyanidins; f. catechins; and g. anthocyanidins to be distinguished.

**Figure 2 antioxidants-12-00539-f002:**
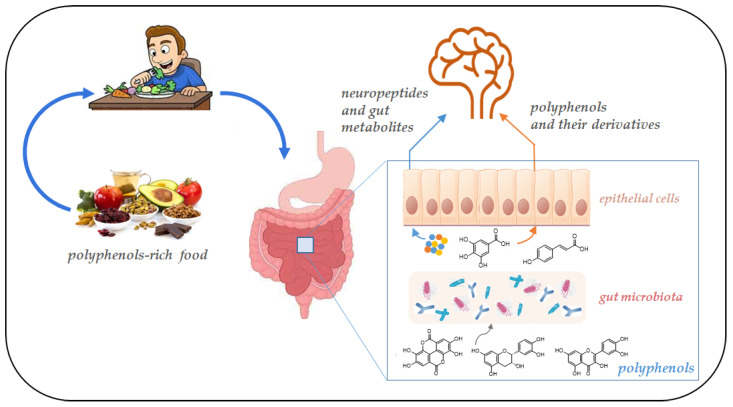
Schematic representation of the gut microbiota–brain interaction. Gut bacteria metabolism produces neurotransmitters and bioactive metabolites from polyphenols. These molecules reach the brain crossing the intestinal barrier.

**Figure 3 antioxidants-12-00539-f003:**
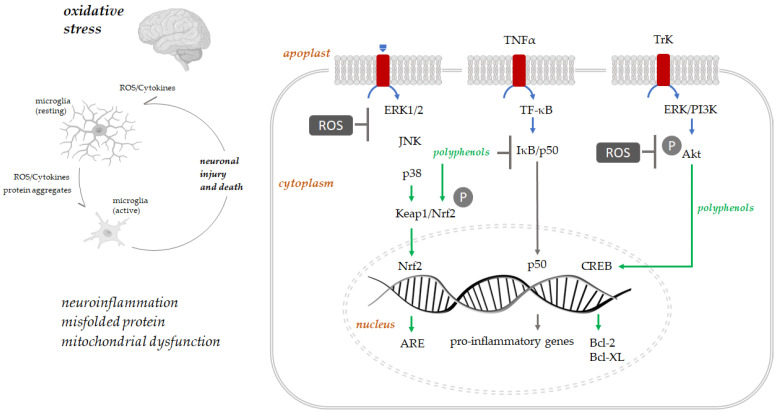
Oxidative stress, neuroinflammation, and neuronal death are strictly associated. The intracellular signaling pathways involved in polyphenols’ neuroprotection are schematized (adapted from [71]).

**Figure 4 antioxidants-12-00539-f004:**
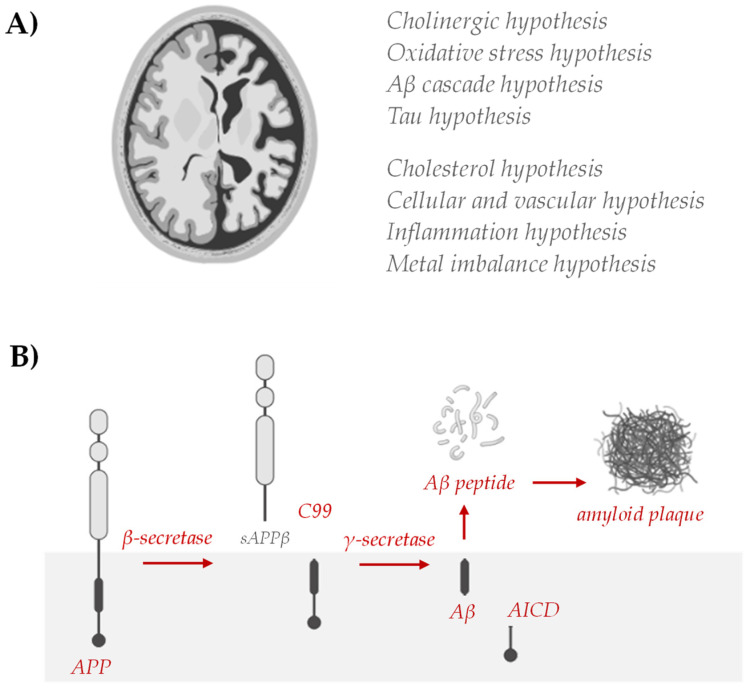
(**A**) Hypotheses of AD include abnormal deposition of Aβ protein in the extracellular spaces of neurons, formation of twisted fibers of tau proteins inside neurons, cholinergic neuronal damage, inflammation, and oxidative stress. Cholesterol has been shown to modulate processes central to the pathogenesis of AD [78], where the vascular hypothesis has been described based on epidemiological studies, as cardiovascular disease contributes to the development and progression of AD and vascular risk factors, such as hypertension, diabetes, and hyperhomocysteinemia, are associated with a significantly higher likelihood of developing AD [79,80]. Evidence that amyloid and tau pathology arises in a constitutively high metal flux environment and that major components of AD pathology may contribute to disease by failing in their metal transport roles has advanced the metal theory in the AD. On the other hand, the discovery of increased levels of inflammatory markers in AD patients and the identification of AD risk genes associated with innate immune functions point to neuroinflammation as a key event [81]. (**B**) The established biochemical alterations of the Aβ cycle, the so-called amyloidogenic pathway, remain the fundamental AD hallmark for targeting therapies.

**Figure 5 antioxidants-12-00539-f005:**
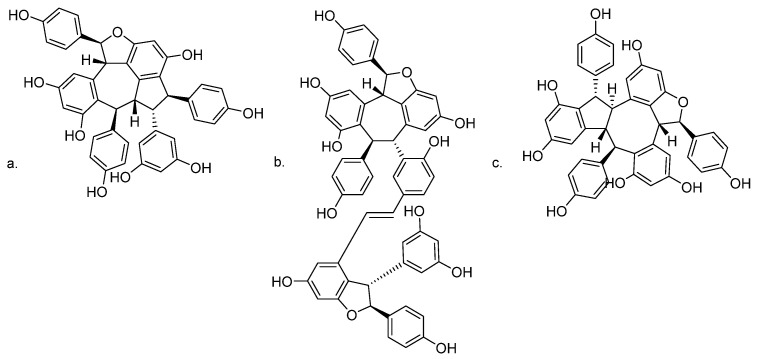
Chemical structures of (**a**) ampelopsin C; (**b**) vitisin A; and (**c**) stenophyllol B.

**Figure 6 antioxidants-12-00539-f006:**
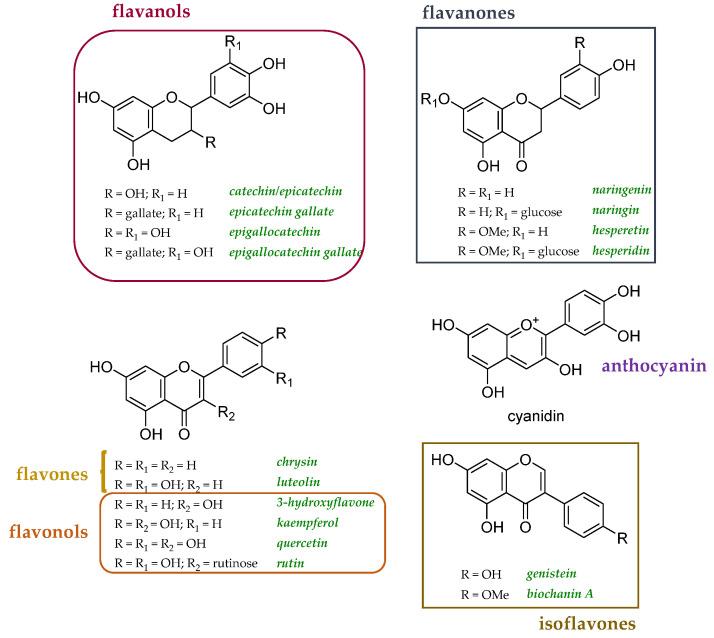
Chemical structures of flavonoids explored as copper transport modulators [125].

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
