# Peer review of "Neuro-Nutraceutical Polyphenols: How Far Are We?"

_antioxidants, 2023, doi:10.3390/antiox12030539_

Round 1

Reviewer 1 Report

Taking into account the fact that polyphenols are” acquirable through diet or nutraceutical supplementation” and targets of their neuroprotective activities are located in the brain, it is required to discuss their bioavailability for dietary exposure (from the alimentary tract) alongside biotransformation and distribution to the brain parent form and metabolites/catabolites. In this context, it would be valuable to present formulation advancement to answer the question, “how far are we?”

Author Response

The authors thank the Reviewer for his/her helpful suggestion. In fact, although the bioavailability of polyphenols and their biotransformation has been introduced in paragraph 3, and different examples have been reported, a further subparagraph (Please see paragraph 5.5 entitled "Polyphenol-based formulations for enhancing their neuroprotective goals", page 19, line 774) has been further created. Furthermore, a conclusion section has been added (page 19, line 815).

Reviewer 2 Report

The manuscript is a summary of the current state of knowledge on the importance of polyphenols in the prevention of neurodegenerative diseases. The Authors indicate a number of mechanisms that may underlie such a prophylactic effects and specific polyphenols that may be subject to individual mechanisms. The review is interesting and should be published after considering several corrections.

The work does not have a summary that would indicate which further research is currently most urgent.

In my opinion, the Authors paid too little attention to phenolic acids, they are mentioned more as human or bacterial metabolites than as important active ingredients of plant-derived foods, for example in relation to cholinesterases inhibition.

Page 6, line 229: Can polyphenols only increase the growth of beneficial microflora, or can they also have an antimicrobial effect in relation to saprophytic microbiota.

Page 10, line 372: The Authors give the number of dementia patients in 2015, is there any data available, what is it like now?

Author Response

1) The review aims at focusing on polyphenol compounds, underlining that, although there is a great literature confusion, polyphenols, from a chemical point of view, are all compounds that share almost two phenolic rings linked to each other by several structural elements (Please see paragraph 2 and references cited therein). Phenolic acids are not polyphenol compounds because their chemical structure is characterized by only one benzene ring. They could be considered, as also detailed in Figure 1, as precursors of polyphenol compounds. The authors thank the reviewer, who provides them the starting point for a further review of the knowledge of simple phenolics and phenolic acids (both hydroxybenzoic acids and derivatives and hydroxycinnamoyl acids and derivatives, as well as their derivatives with C6C2 backbone).

2) We thank the Reviewer for his/her question. Many studies show that polyphenols, or their metabolites from microbial enzymatic activity, possess antimicrobial activity against Gram-negative and Gram-positive bacteria.  However, this antimicrobial activity is described to promote the healthy balance of the intestinal microflora. Two sentences and one reference have been added in the text to better explain this concept (lines 229-233 and 235-238).

Reference added:

Dong R., Shuai Liu, Jianhua Xie, Yi Chen, Yuting Zheng, Xingjie Zhang, En Zhao, Zipei Wang, Hongyan Xu, Qiang Yu. The recovery, catabolism, and potential bioactivity of polyphenols from carrots subjected to in vitro simulated digestion and colonic fermentation. Food Res Int. 2021 May;143:110263.

3) The authors  have been updated the on the basis of WHO report and accordingly changed the manuscript (page 9, lines 370-374)

Round 2

Reviewer 2 Report

In its current version, the manuscript can be accepted for publication.